



# Characterizing lead-rich particles in Beijing's atmosphere following coal-to-gas conversion: Insights from single particle aerosol mass spectrometry

Xiufeng Lian[1, 2], Yongjiang Xu[1], Fengxian Liu[3], Long Peng[4], Xiaodong Hu[5], Guigang Tang[6], Xu Dao[6], Hui Guo[7], Liwei Wang[8], Bo Huang[2], Chunlei Cheng[1], Lei Li[1], Guohua Zhang[9], Xinhui Bi[9], Xiaofei Wang[10], Zhen Zhou[1], Mei Li[1,*]

[1] College of Environment and Climate, Institute of Mass Spectrometry and Atmospheric Environment, Guangdong Provincial Engineering Research Center for On-line Source Apportionment System of Air Pollution, Jinan University, Guangzhou , 510632 PR China

[2] Guangzhou Hexin Instrument Co., Ltd., Guangzhou 510530, Guangdong, China

[3] Taiyuan University of Technology, Taiyuan 030024, PR China

[4] College of Ecology and Environment, Xin Jiang University, Urumqi 830046, PR China

[5] Jiangmen Laboratory of Carbon Science and Technology, Hong Kong University of Science and Technology (Guangzhou), Jiangmen 529100, PR China

[6] China National Environmental Monitoring Centre, Beijing 100012, PR China

[7] Hunan Province Environmental Monitoring Center, Changsha 410014, PR China

[8] Environment Emergency Monitoring and Accident Investigation Center, Jiaxing 314000, PR China

[9] State Key Laboratory of Organic Geochemistry and Guangdong Provincial Key Laboratory of Environmental Protection and Resources Utilization, Guangzhou Institute of Geochemistry, Chinese Academy of Sciences, Guangzhou 510640, PR China

[10] Department of Environmental Science and Engineering, Shanghai Key Laboratory of Atmospheric Particle Pollution and Prevention, Fudan University, Shanghai 200433, China

*Correspondence to: limei@jnu.edu.cn*

**Abstract.** Coal-to-gas (CTG) policies are important energy transformation strategies to address air pollution issues, but how well it improves atmospheric Lead (Pb) pollution remains poorly understood. By the end of 2018, Beijing had achieved coal-free in urban and plain areas. The mixing state and atmospheric chemical processes of Pb-rich particles in Beijing were monitored using a single particle aerosol mass spectrometry (SPAMS) during 2019. Based on a large dataset of mass spectra, this study find that the number fractions of Pb-rich particles, as well as two types of Pb-rich particles (K-Na-EC and K-OC) related to coal combustion during heating period, show lower than those after the heating period. Based on concentration-weighted trajectory plots, the results indicate that lead aerosols mainly derive from the transmission of surrounding provinces.

Lead nitrate is one of the important forms of lead in aerosol particles, most contributed by photo-chemical reactions in spring, fall, and winter. Due to the decomposition of nitrate during high temperatures, the aqueous reactions mechanism contributes more to lead nitrate in summer. These results improve our understanding of the seasonal distribution, formation mechanisms, and influencing factors of toxic Pb-containing particles after CTG.



## 1 Introduction

Lead (Pb), as one of the ubiquitous toxic metals, has been widely noted for its serious harm to human health (Cho et al., 2011; Grandjean and Herz, 2015; Lu et al., 2019; Schindler et al., 2021; Sommar et al., 2013). Pb-containing particles enter the lungs and blood of the human body through the respiratory tract, and accumulate in circulation within the body using blood as a carrier (Oberdörster et al., 2004); thus, they can severely damage to DNA, kidney, and nervous system (Gu et al., 2018; Shtepliuk et al., 2018; Wani et al., 2015; Yedjou et al., 2016). Previous studies have suggested that no level of Pb in the human

body is considered safe (Jakubowski, 2011; Rossi, 2008). Additionally, atmospheric Pb is also considered to contribute to climate modification as a cloud condensation nucleus (Cziczo et al., 2009; Ebert et al., 2011).

Pb enters the atmosphere through emissions from natural and anthropogenic sources. The Pb emission fluxes from anthropogenic sources are at least $1-2$ orders of magnitude higher than those from natural sources (Komárek et al., 2008). Since the 1980s, atmospheric Pb has declined sharply due to leaded gasoline being phased out (Cho et al., 2011; Kristensen,

2015). Later, the decrease in airborne Pb emissions from automobiles increased the relative contribution of smelters, coal combustion, and waste incineration (Cai et al., 2017; Cho et al., 2011; Liang and Mao, 2015; Lu et al., 2019; Peng et al., 2020; Zhang et al., 2009; Zhao et al., 2017). Atmospheric Pb is primarily released during high-temperature processes due to condensation of its vapors and/or coagulation with existing particles (Csavina et al., 2014; Murphy et al., 2007). Lead production smelting processes and coal combustion processes emit PbS, PbO, and $PbSO_4$ into the atmosphere (Koukouzas et

al., 2011; Sobanska et al., 1999; Tian et al., 2013; Yoshiie et al., 2013). The other two forms of lead compounds are $PbCl_2$ and $Pb(NO_3)_2$, which mainly come from the atmospheric aging process of Pb-containing aerosols (Schindler et al., 2021). $PbCl_2$ formed through the heterogeneous reaction between PbO and HCl during smelting combustion (Furimsky, 2000; Ohmsen, 2001) or waste incineration (Li et al., 2017; Moffet et al., 2008; Zhang et al., 2009). $Pb(NO_3)_2$ generate from the chemical transformation of PbO and $PbCl_2$ in the atmospheric process (Moffet et al., 2008; Peng et al., 2020). These processes are crucial

for Pb showed a significant negative impact on health because of the high solubility of $Pb(NO_3)_2$ (Bas et al., 2016; Bas and Kalender, 2016).

Coal combustion during winter heating is an important source of Pb-containing particles in Beijing, China (Li et al., 2012; Murphy et al., 2007; Peng et al., 2020). Since 2013, the government introduced a series of strict measures to advance the conversion of coal to clean energy. By the end of 2018, villages in Beijing's plain area have achieved coal-free transformation

(Wang et al., 2021). Urban clean fuel centralized heating has been completed. Therefore, the elimination of coal combustion thus provided a valuable opportunity to fully understand the contribution of coal combustion sources to atmospheric Pb and the atmospheric Pb sources previously masked by coal combustion sources. However, there is still a lack of research on the source, transport, and evolution process of Pb particles after CTG.



Traditional atmospheric Pb data are off-line collected by Particle Analysis by Laser Mass Spectrometry (PALMS),

Transmission Electron Microscope (TEM), Scanning Electronic Microscopy (SEM) (Murphy et al., 2007; Schindler et al.,

2021). Off-line techniques are useful for providing accurate mass concentration values for Pb, but show poor temporal

resolution. More recently, mass spectrometry-based online techniques such as single-particle mass spectrometer (SPMS)(Lu

et al., 2019) have been favoured. Although these techniques are typically not quantitative, they provide information on the

sources and mixing state of Pb in real time. This information is critical to understanding the atmospheric transport and aging

processes of Pb. Lu et al. (2019) analyzed the mixed state of Pb-containing particles monitored by SPMS and found that the

sources of particulate Pb in Guangzhou, China include coal combustion, waste incineration, vehicle exhaust, industrial process,

and dust. Moffet et al. (2008) used a SPMS to monitor the diurnal variation characteristics of Zn or Pb single particles in

industrial areas in Mexico, and found that Zn or Pb chloride convert to metal nitrate began each day at ~7 am.

To better understand the mixing state, source, and atmospheric evolution process of atmospheric Pb after CTG, we investigate

Pb-containing individual particles with a high time resolution in the atmosphere of urban Beijing from spring to winter in 2019.

The three specific goals are to: 1) compare the differences of Pb-rich particles between heating and non-heating periods; 2)

investigate the sources and mixing state of Pb-rich particles after CTG; and 3) expound the diurnal and seasonal variation

characteristics of particulate lead nitrate and its possible formation mechanism of atmospheric Pb after CTG.

## 2 Field measurements

### 2.1 Data Collection

The sampling site was located in Beijing at the China National Environmental Monitoring Centre (40.05° N, 116.43° E; 40 m

a.s.l.) in Beijing. Sampling was conducted during spring (March 9–April 8, 2019), summer (June 16–July 9, 2019), fall

(September 15–October 16, 2019), winter 2019 (December 15, 2019–January 15, 2020). The central heating period in Beijing

is from 00:00 on November 15th to 23:00 on March 15th. Sampling in spring and winter covers a central heating period,

respectively. Local meteorological parameters, including RH, temperature (T), wind speed, wind direction, and the

concentrations of $NO_2$, $SO_2$, $O_3$, and $PM_{2.5}$ from 2019 to 2020 are provided by the China National Environmental Monitoring

Centre (**Table S1**).

This study used a single particle aerosol mass spectrometer (SPAMS, Hexin Analytical Instrument Co., Ltd., Guangzhou,

China) to monitor the size and chemical composition of aerosol particles with a vacuum aerodynamic diameter of 0.1–2.0

micrometers ($d_{va}$). Ambient particles are separated by a $PM_{2.5}$ cyclone separator and dried with silica gel before entering the

single particle aerosol mass spectrometer. The detailed setup and mechanisms of the SPAMS have been described elsewhere

(Li et al., 2011). Briefly, aerosol particles were introduced into the SPAMS through a 0.1 mm critical orifice at a flow rate of

80 ml min$^{-1}$. Then, they were focused and aerodynamically sized by two continuous diode Nd:YAG laser beams (532 nm),





followed by desorption/ionization by a pulsed laser (266 nm) that was triggered exactly based on the time of flight during the

sizing. The positive and negative fragments generated were then detected using a time-of-flight spectrometer.

**2.2 Data Analysis**

Data analysis is performed by importing single-particle size and mass spectra into MATLAB (The MathWorks, Inc.) using the

Continuation Core (COCO; version 3.0) toolkit. A total of ~650 000 to ~3 400 000 particles with both positive and negative

ion mass spectra were obtained by the SPAMS. Among them, around 20 000 to 140 000 (2.8−4.4%) Pb-containing particles

were identified with ion signals > 5 (> 3 times the noise level) at mass-to-charge ratios ($m/z$) of 206, 207 and 208. Pb-rich

particles are defined as particle with a relative intensity of $^{208}$Pb greater than 5% of the total intensity in a mass spectrum,

which is the same criterion that has been used in previous online Pb single-particle studies (Cai et al., 2017; Peng et al., 2020;

Zhang et al., 2009). We measured 8 034−23 941 Pb-rich particles, accounting for 0.7−1.2% of all detected particles and

16.0−44.5% of Pb-containing particles (**Table 1**). During the sampling period, there a significant correlation between the

number of Pb-containing particles and Pb-rich particles in all the detected particles, with correlation coefficients *r > 0.85* and

*p<0.01*. Moreover, the size distribution characteristics of Pb-containing particles and Pb-rich particles are similar (**Fig. S1**).

The correlation between Pb-containing particles and Pb-rich particles in winter (*r = 0.47, p<0.01*) is much lower than that in

other seasons.The number fractions of Pb-rich particles decreased significantly, accounting for only 16% of Pb-containing

particles in winter, which is 40% lower than that in other seasons. Previous results showed that Pb-rich particles typically

undergo less aging process than non Pb-rich particles (Zhang et al., 2009). Thus the weaker correlation in winter may be related

to the accumulation of non Pb-rich particles after the conversion of Pb-rich particles to non Pb-rich particles caused by poor

diffusion conditions. Therefore, this study focuses on Pb-rich particles with higher relative strength fractions of Pb than that

in Pb-containing particles, which are more likely to imply higher lead concentrations in individual particles and had relatively

simple chemical histories.

The screened Pb-rich particles were subsequently analyzed using an adaptive resonance theory-based neural network algorithm

(ART-2a) (Song et al., 1999), with a vigilance factor of 0.75, a learning rate of 0.05, and 20 iterations. To analyze the source

of Pb-rich particles, approximately 95% of the particles were classified by ART-2a and manually classified into six types based

on positive spectral signal intensity. They are named potassium-sodium (K-Na), K-Na internally mixed with Fe (K-Na-Fe), K-

Na internally mixed with Cu (K-Na-Cu), K-Na internally mixed with Zn (K-Na-Zn), K-Na internally mixed with elemental

carbon (K-Na-EC), and potassium internally mixed organic carbon (K-OC). The remaining particles were classified as

"Others". The average mass spectra for the main Pb-rich particle types are shown in **Fig. 1**.

The definition of lead nitrate (Pb-N) particles in this study is the particles mainly composed of nitrate in the negative spectrum





of Pb-containing particles, and the nitrates peak area is strongly correlated with lead (Peng et al., 2020). In this study, the

average mass spectrum and digital spectrum of Pb-N particles are shown in **Fig. 2** and **Fig. S2−3**. Pb-N particles mainly consist

of nitrate in the negative spectrum, and iron, sodium, potassium, and a small amount of organic compounds in the positive

spectrum. There are significant correlation (*r = 0.82−0.98, P < 0.01*) between the total peak area of nitrates (sum peak area of

*m/z* −46 and −62) and the total peak area of lead (sum peak area of *m/z* 206, 207, and 208) in Pb-N particles(**Fig. S4**). The

weak correlation between nitrates and sodium, potassium and iron (*r = 0−0.42, P < 0.01*) are also found. Thus, these Pb-N

type particles are also most likely represented as $Pb(NO_3)_2$ in this study.

**2.3 Concentration Weighted Trajectory (CWT) model**

The CWT model was applied to localize potential source areas affecting aerosol particles at a receptor site (Watson et al.,

2008). In this procedure (Equation R1), each grid cell gets a weighted concentration obtained by averaging sample

concentrations that have associated trajectories that crossed that grid cell as follows (Hsu et al., 2003):

$$C_{ij} = \frac{1}{\sum_{l=1}^{M} \tau_{ijl}} \sum_{l=1}^{M} C_l \, \tau_{ijl} \, ) \tag{R1}$$

$$W(i,j) = \begin{cases} 1.0 & (3n_{ave} < n_{ij}) \\ 0.7 & (1.5n_{ave} < n_{ij} < 3n_{ave}) \\ 0.4 & (n_{ave} < n_{ij} < 1.5n_{ave}) \\ 0.2 & (n_{ave} < n_{ij}) \end{cases} \tag{R2}$$

$C_{ij}$ is the weighted average concentration in a grid cell (i, j), $l$ is the index of the trajectory, M is the total number of trajectories,

$C_l$ is the concentration observed on the arrival of trajectory $l$, $\tau_{ijl}$ is the time spent in the $ij^{th}$ cell by trajectory $l$. $W(i,j)$ is the

weighting function and $n_{ij}$ is the number of endpoints through the $ij$ cell. The weighted concentration fields show concentration

gradients across potential sources. The back-trajectory analyses were conducted with the HYSPLIT4 model from the U.S.

National Oceanic and Atmospheric Administration (Draxler and Rolph, 2003), with the arrival altitude and calculation time

set at 100 m and 72 h, respectively.

**3 Results and discussions**

**3.1 Mixing state and sources of Pb-rich particles**

K-Na-EC particles can be distinguished from the other Pb-rich particles by their distinct EC cluster ions with $C_n^+/C_n^-$ (n: 2–5)

in the mass spectra (**Fig. 1**), accounting for 1.5–8.9% of all the detected particles (**Table 1**). The spectral characteristics of K-

Na-EC particles in this study are similar to those of Pb-EC/OCEC-rich particles in previous studies of Pb-containing single

particles, which are confirmed to be most likely attributed to combustion emissions, especially from coal combustions (Cai et

al., 2017; Peng et al., 2020). The number fractions of Pb-EC particles during the heating period was ~5−10%, while before the

heating period it was ~1−2% in the winter of 2014 (Peng et al., 2020). EC particles are directly emitted from sources such as

vehicle exhaust, industrial processes, and biomass burning due to incomplete combustion of organic species (Tiwari et al.,

2013). Pb-containing particles from vehicle combustion are often coated with semi-volatile organic carbon species, and K-Na-



EC from diesel truck exhaust mostly coexists with phosphate (Amann and Siegla, 1982; Lu et al., 2019). In this study, K-Na-EC particles contain very weak OC fragments such as $m/z$ 51 [$C_4H_3$]$^+$ and 63 [$C_5H_3$]$^+$), and the number fractions of phosphate ($m/z$ –79 [$PO_3$]$^-$) in K-Na-EC particles accounted for only 10.4%. In addition, Pb is only a minor component in particles from biomass burning (Bi et al., 2011; Hudson et al., 2004; Murphy et al., 2007). Considering these factors, the K-Na-EC particles in this study are most likely from coal combustion.

OC particles exhibit spectral variation with strong $m/z$ 39 [K]$^+$and weak OC fragments ($m/z$ 37 [$C_3H$]$^+$ and 51 [$C_4H_3$]$^+$), accounting for 1.0–4.4% of all the detected particles. The significant correlation between K-OC and K-Na-EC particles ($r = 0.58, p < 0.01$; **Fig. S5(a)**) indicates that they may be homologous (Cai et al., 2017; Lu et al., 2019). The K-Na-EC and KOC particles showed similar distribution that the number fractions decreased with increasing $d_{va}$, and represent the highest fraction (46.4–71.4%) in the size range of 0.1–0.4 µm (**Fig. S6**). Additionally, K-Na-Cu particles show similar size distribution characteristics with K-OC and K-Na-EC particles, accounting for 1.2–3.3% of all the detected particles. K-Na-Cu particles are identified by the intense ion signals from Cu (m/z 63/65 [Cu]$^+$) and phosphate, with a peak particle size distribution of 0.5–0.6 µm. They are probably from industrial coal combustion, as the mass spectrum and particle size distribution characteristics of K-Na-Cu particles are similar to those of Pb-containing particles released from industrial coal combustion (Lu et al., 2019).

K-Na-Fe particles exhibit spectral variation with strong a dominant iron signal at $m/z$ 56 [Fe]$^+$ and a weaker signal from its natural isotope $m/z$ 54 [Fe]$^+$ in the positive spectrum. K-Na-Zn is recognized by a strong signal at $m/z$ 64 [Zn]$^+$ and a weaker signal from its natural isotope $m/z$ 66 or 68 [Zn]$^+$ in the positive spectrum. Unlike the particle types mentioned earlier, the negative spectra of K-Na-Fe and K-Na-Zn particles contain strong nitrate ($m/z$ –46 [$NO_2$]$^-$ and –62 [$NO_3$]$^-$) peaks, but the sulfate ($m/z$ –97 [$HSO_4$]$^-$) peaks are very weak. They are distributed in larger particle sizes (>0.5 µm), indicating that they are more aged than those from combustion. K-Na-Fe and K-Na-Zn particles account for 4.9–6.8% and 0.1–0.9% of all the detected particles, respectively. K-Na-Fe and K-Na-Zn particles are highly likely to originate from industrial process, such as metal smelting processes. Industrial emissions of Pb are mainly from nonferrous metal smelting, usually mixed with Fe and/or Zn in the particle phase (Batonneau et al., 2004; Lu et al., 2019; Moffet et al., 2008; Tian et al., 2015). Lu et al., (2019) demonstrated that K-Na-Zn particles in environmental aerosols are most likely to originate from industrial processes by comparing atmospheric LCP mass spectra with authentic Pb emission source mass spectra. Previous research has shown that the number fractions of the Pb-rich particles mixed with Fe and Zn significantly decreases during the heating season, with decreases of 8% and 6%, respectively (Peng et al., 2020).

K-Na particles exhibit spectral variation with strong $m/z$ 39 [K]$^+$and $m/z$ 23 [Na]$^+$ in the positive spectrum, and strong nitrate in the negative spectrum. K-Na particles are the most abundant particle type in environmental aerosols, accounting for 78.4–84.3% of all detected particles. The vast majority of K-Na particles are emitted from non-coal combustion sources (Murphy et al., 2007; Tian et al., 2015). K-Na particles are not correlated with the number fractions of K-Na-EC and K-OC particles. The relative number fractions of K-Na particles decreased significantly in the heating season. Particles of K-Na mixed with Li ($m/z$ 7 [Li]$^+$), which may come from coal combustion fly ash (Furutani et al., 2011; Guazzotti and S., 2003; Liu et al., 1997), account for only 3.5% (by number) of all K-Na particles. Most (83.1–97.8%) of K-Na particles are distributed in the particle size larger than 0.5 µm, same as K-Na-Zn and K-Na-Fe particles. Significant correlation ($r = 0.61, p < 0.01;$ Fig.S7(b)) between the number fractions of K-Na and K-Na-Fe is also observed. These results suggest that K-Na and K-Na-Fe particles have the same primary sources and undergo similar atmospheric aging processes. We speculate the major primary sources of Pb in Beijing have changed from combustion processes and the iron/steel industry after CTG.



### 3.2 The reduced contribution of local coal combustion to Pb-rich particles after coal-to-gas

The variation characteristics of Pb-rich particles and various types of Pb-rich particles during heating period (HP) and after HP in spring 2019 are shown in **Fig. 3**. The average number fractions of Pb-rich particles during HP is 28.6% lower than that after HP, and this trend of lead nitrate particles (42.2%) is more significant (**Fig. 3(b)**). The maximum hourly number fractions of Pb-rich and Pb-N particles after HP are usually one to two twice that during HP (**Fig. 3(a)**). This is a significant decrease compared to doubled the average and maximum hourly number fractions of Pb-rich particles during the past coal-fired centralized heating period in Beijing (Peng et al., 2020). In addition, during the heating period, the concentration of pollutants ($PM_{2.5}$, $NO_2$, $SO_2$, CO, and $O_3$) are also lower than those during non-heating periods (**Fig. S7(a)**). The relative peak area of Pb in Pb-rich particles in heating season including spring and winter are about 15% lower than that in non-heating season (**Fig. S7(b)**). Previous studies on the impact of coal-to-gas conversion policies on air quality have shown that this policy led to $PM_{2.5}$, $SO_2$, and other air quality parameters significantly improved. On average, the "coal-to-gas" policy reduced $SO_2$, $NO_2$, $PM_{10}$, $PM_{2.5}$, and CO by 12.08%, 4.89%, 13.07%, 11.94% and 11.10% per year from 2014 to 2018 in Beijing, respectively (Liu et al., 2020). Our results provide evidence that the "coal-to-gas" policy has also effectively improved the pollution of Pb-containing aerosols.

We note that the total relative number fraction of K-Na-EC and KOC particles associated with coal combustion sources doubled during the HP compared to the non-heating period (**Fig. 3 (c)**). And in winter, the total relative number fraction of K-Na-EC and KOC particles is more than twice that of non heating seasons in summer and fall (**Fig. S8**). Sampling in winter is 25 days longer than in spring during the heating period, and the total relative quantity fraction of K-Na-EC and KOC particles is 60% higher than in spring. To identify the potential sources for the Pb-rich particles after CTG, the concentration-weighted trajectory (CWT) plots from spring to winter are analyzed (**Fig. 4**). The number of high weighted cells in the web version presented the potential major original source areas. The potential source area of atmospheric Pb is mainly dominated by external transport and exhibits seasonal variation characteristics. Pb-rich particles mainly contributed by eastern Inner Mongolia, the central region of Liaoning Province, and at the junction of Anhui Province, Shandong Province, and Henan Province during spring. The high-value areas during summer and fall are mainly distributed in the eastern part of Shandong Province and Tianjin, as well as the western part of Liaoning Province. The potential winter source areas are distributed in the long-distance transmission of air masses from eastern Mongolia and the northern regions of Tianjin and Hebei Province. Previous research results have shown that on average 45% of Pb in $PM_{2.5}$ in urban Beijing was transported (Cai et al., 2017). It is noticeable that lead smelters and metal refining plants are wildly distributed in this area with rich mineral resources (Cai et al., 2017; Liu et al., 2018; Tian et al., 2015). These regions are also the regions with higher lead emissions in China in 2020 (Tong et al., 2024).

### 3.3 Characteristics of size and mixing state during HP and after HP after coal-to-gas

The mixed state with anions is an important fundamental data for further understanding the form of lead in environmental aerosols. In this study, about 60% of the Pb-rich particles are mixed with chloride and oxygen, and this proportion was similar during the heating period and non-heating period in spring, and heating period in winter (**Fig. 5**). The mixing ratio of Pb-rich particles and sulfate particles during the spring heating period is about 10% higher than that during the non-heating Pb-rich particles, which may be affected by the increase in sulfur dioxide concentration during the heating period (**Table S1**). After coal-to-gas conversion, the mixing state of lead in spring and summer are more complex than in summer and winter, with lead and sulfate and chloride salts being twice as much in spring and winter as in summer and autumn.

Addtionally, previous research has shown that the main forms of lead in aerosol particles include lead oxide, lead chloride, lead sulfate, and lead nitrate (Cai et al., 2017; Lu et al., 2019). In this study, almost all lead are mixed with nitrate, which is



much higher than its mixing with sulfate, chlorine, and oxygen. As shown in **Fig. S9**, Pb-N particles increase with the increase

of particle size, with more than 85% of particles distributed between 0.5 and 1.0 μm, indicating that most of the lead nitrate comes from the secondary generation of Pb-containing particles during transportation. The number fractions of Pb-N particles in Pb-rich particles are 17.1−39.1% from spring to winter (**Table 1**). Among them, the number fractions of Pb-N particles are higher in summer and fall than those in spring and winter. However, this result is inconsistent with the distribution characteristics of high nitrate levels in winter and lower nitrate levels summer. To unravel this mystery, we further explore the

possible formation mechanism of lead nitrate in next section.

### 3.4 The possible formation mechanism of $Pb(NO_3)_2$ in the particle after coal-to-gas

Except for summer, the number fractions of Pb-N particles are higher during the day and lower at night, with high values mainly distributed between 10:00 and 15:00 (**Fig. 6**). In spring, Pb-N particles account for the lowest amount of all detected particles from 22:00 to 00:00, with an average of about 0.1%. After 00:00, the number fractions of Pb-N particles increase

gradually until 9:00−15:00, and the number fractions of Pb-N particles increase to 0.3%, and then slowly decrease. The peak number fractions of Pb-N particles in fall lags behind that in spring by about two hours. It starts to rise rapidly at 10:00 in the morning and rises from 0.3% to 0.8% in three hours. After 14:00, it slowly drops to ~0.3% at 21:00, fluctuates slightly between 0.2−0.3% from 22:00 to 10:00 the next day. The number fractions of Pb-N particles in winter with time is similar to that during fall, but the difference between the high value in daytime and other times is small, only 0.07%. In Mexico City aerosols, the

conversion of lead chloride to lead nitrate occurs between 7:00−21:00 and peaks at 12:00−2:00 due to effective formation of gaseous $HNO_3$ after sunrise (Moffet et al., 2008). Furthermore, the number fractions of Pb-N particles are slightly higher on sunny days than on non-sunny days in spring and winter (**Fig. 7**). Ozone, which reflects the activity of photochemical reactions, is significantly higher on sunny days than on non-sunny days, with an average value of 1.7−71.0% higher from spring to winter (**Fig. S10**). In particular, the low number fractions (~0.2%) of Pb-N particles is from 12:00 to 23:00, and the high value (~3.5%)

is from 00:00 to 11:00 in summer. Previous results have shown the $Pb(NO_3)_2$ did not show any significant diurnal variation, both daytime and nighttime atmospheric processes contributed to the observed $Pb(NO_3)_2$ (Peng et al., 2020). These results support the importance of photo-oxidation for the formation of nitric acid and lead nitrate, and highlight the seasonal differences in the formation process of lead nitrate.

Previous studies revealed that $NO_2$, RH, and T are key factors affecting the formation of lead nitrate in atmospheric aerosols.

A significant positive correlation ($r = 0.88$) between the RH of nitrate and $NO_2$ was observed (Peng et al., 2020). Lead nitrate abundance begins to increase at sunrise and peaks at 12:00−2:00 (Moffet et al., 2008), consistent with the diurnal variation in temperature. In this study, the number fractions of Pb-N particles increased with the increase of temperature between −10 ℃ and 30 ℃, and decreased with the increase of temperature above 30 ℃ (**Fig. 8**). Solar radiation affects the formation of OH radicals, and OH radicals are oxidants that transform $NO_2$ into $HNO_3$, so a high concentration of OH radicals is conducive to

the formation of $HNO_3$. At the same time, higher temperature is also conducive to $NO_2$ and OH reaction (Slater et al., 2019). Specifically, the number fractions of Pb-N particles decreases with increasing T, especially when the temperature exceeds 30 ℃, which may be an indirect result of the decomposition of nitrate particles caused by high temperature (Song et al., 2019; Luo et al., 2020).

With the increase of $NO_2$, the number fractions of Pb-N particles decrease significantly in summer, but did not change

significantly in other seasons. The effect of $NO_2$ on the formation of lead nitrate is limited by RH. The relative intensity of nitrate exceed 12% in most situations when the RH was higher than 60%, while the relative intensity of nitrate reached a maximum of 21% with RH above 80% (Peng et al., 2020). The decreased number fractions of Pb-N particles slightly with the increase of RH. This is inconsistent with the result that high RH promotes the formation of lead nitrate(Peng et al., 2020).



It is worth noting that this does not indicate that high RH is not conducive to the formation of lead nitrate, but may be influenced by the photo-chemical oxidation reaction promoted by high T and low RH during the day than the aqueous reaction promoted by high humidity on the formation of lead nitrate.

Lead nitrates in atmospheric aerosols are generated by heterogeneous reactions of lead chloride or lead oxide with $NO_2$ or by aqueous reactions with nitric acid (Moffet et al., 2008; Peng et al.,2020). Primary emissions of $PbCl_2$ particles underwent heterogeneous transformations into nitrate particles as soon as photochemical production of nitric acid (Moffet et al., 2008).

Peng et al. (2020) used a SPAMS to study the transport and aging mechanism of lead in the winter atmosphere of Beijing before "coal-to-gas". The study included heating and non heating periods, and the results emphasized the importance of heterogeneous hydrolysis of during heating period before "coal-to-gas" on the formation of $Pb(NO_3)_2$. The results of this study show that after the implementation of the coal-to-gas policy, the formation of lead nitrate in aerosols may shift from aqueous reaction dominance to photochemical reaction dominance in winter.

### 3.5 Uncertainties and Limitations

Although promising results and good comparisons are obtained from the analysis of particle size, particle number and mixing state monitored by SPAMS, there are some uncertainties and limitations in this study. First, due to its lower detection efficiency at smaller sizes, we may underestimate Pb abundance on small particles. Secondly, we overlook the matrix effect, which affects the collection efficiency of single particle aerosol mass spectrometry for different components. Regardless of the possible

uncertainties mentioned above, the agreement between the numbers of all the detected particles and $PM_{2.5}$ concentrations agreed well with each other($r = 0.58-0.70; p < 0.01$), suggesting no strong impacts due to these uncertainties.

A limitation or bias of this study is the use of RPA spectral data in these qualitative individual masses, rather than absolute mass concentrations of lead. It should be noted that due to various uncertainties caused by laser ionization, such as matrix effects and incomplete ionization, laser ablative based SPAMS are still very challenging in providing quantitative information (Jeong et al., 2011; Healy et al., 2013; Zhou et al., 2016). Although different at the individual particle level, RPA is still a good

indicator for studying the atmospheric processing and mixing state of various particle types, including Pb-containing particles (Moffet et al., 2008; Zauscher Et al., 2013; Peng et al., 2020). In the present study, although it is not enough to quantitatively evaluate the effect of CTG policies on the improvement of Pb in aerosol, our results do successfully identify a distinct behavior of Pb-rich particles in the ambient atmosphere, and thus have the advantage of qualitative or semi-quantitative understanding of the evolution of Pb-rich particles after coal-to-gas transformation. Future work should target quantitative understanding of

Pb concentration variations in aerosols and atmospheric evolution processes of Pb such as photo-oxidation.

Another limitation is the short sampling time (6 days) during the spring heating period, which may cause deviations in the mean value and assessment of the number fractions of Pb-rich particles during the heating period, resulting in poor data representation. However, the physical and chemical characteristics such as particle type and mixing state of Pb-rich

particles during heating periods in spring or winter are similar (Fig. 5 and Fig. S8), indicating that the short sampling time in spring heating season does not significantly affect the accuracy of data results. Future work could extend the sampling time during heating to make the results more representative.

### 4 Conclusions and implications

We studied the sources, mixing states, and seasonal variations of lead particles in the urban atmosphere of Beijing after coal-to-gas conversion, with a focus on the formation mechanism of toxic lead nitrate particles. K-Na-EC and K-OC particles are associated with coal combustion, and K-Na, K-Na-Fe, K-Na-Zn particles tend to come from iron/steel industries. Our results show that the contribution of coal combustion to atmospheric Pb decreases significantly after CTG, and the iron/steel industries



becomes the most important emission source. The atmospheric Pb in Beijing urban area is mainly transmitted from surrounding

provinces such as Mongolia Province, Hebei Province, Liaoning Province, Anhui Province and Shandong Province. Almost

all lead are internally mixed with nitrate, which is much higher than its mixing with sulfate, chlorine, and oxygen. The number

fractions of Pb-N particles in Pb-rich particles are $17.1-39.1\%$ from spring to winter, which is about twice as high in summer

and fall as in spring and winter. The diurnal variation of number fractions of Pb-N particles show obvious seasonal difference.

Results from this study show that the gas phase photo-oxidation contributed a lot to the formation of lead nitrate particles,

especially in spring and fall.

Our results provide observational evidence for the effect of CTG policy on the reduction of Pb. This broadens the understanding

of CTG to improve air quality, provides a scientific basis for a more accurate assessment of the comprehensive effect of CTG

policies on air quality improvement (Liu et al., 2020). Since there is considerable debate regarding the atmospheric chemistry

of Pb (Moffet et al., 2008; Peng et al., 2020), our findings further emphasize the importance of photochemical reactions and

further clarify their seasonal differences in the formation process of lead nitrate. Consider that a series of CTG policies have

been implemented in Beijing ,Tianjin, and 26 other cities in Hebei, Shanxi, Shandong, and Henan provinces)

(https://www.gov.cn/xinwen/2017-12/20/content_5248855.htm), the research methods and conclusions of this study can also

be extended to these cities.



*Supplement.* Supporting information includes ten figures (Fig. S1-S10), and one table (Table S1) related to the manuscript.


*Data availability.* The observational data including SPAMS and meteorological parameters used in this study are available from corresponding authors upon request (limei@jnu.edu.cn).

*Author contributions.* ML design the research with input from XW, XB, GZ, and ZZ. YX, GT, and XD collected samples. XL

processed data and wrote the manuscript. FL, PP, XH, HG, and LW had an active role in supporting the sampling work. BH, CC, and LL had an active role in supporting the writing-original work. All authors contributed to the discussions of the results and refinement of the manuscript.

*Competing interests.* The authors declare that they have no conflict of interest.


*Financial support.* This study is supported by the National key research and development program (2023YFC3705502), the National Natural Science Foundation of China (Grant No. 42407355), Special Fund Project for Science and Technology Innovation Strategy of Guangdong Province (Grant No. 2019B121205004), Guangdong Foundation for Program of Science and Technology Research (Grant No. 2020B1212060053), State Environmental Protection Key Laboratory of Monitoring for

Heavy Metal Pollutants (Grant No. SKLMHM202325).



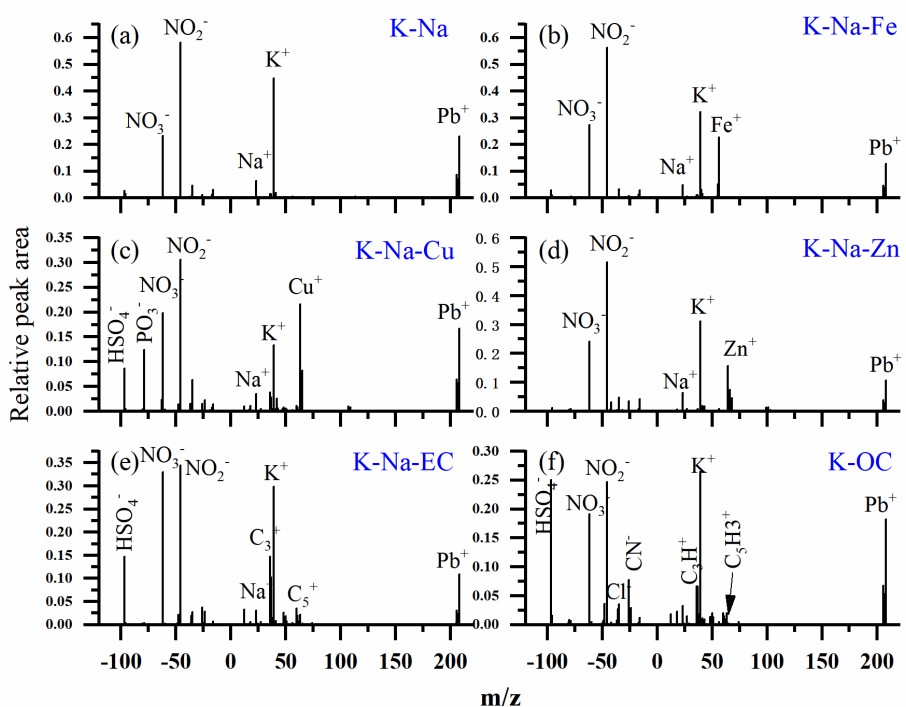

**Figure 1. Average mass spectra for the main Pb-rich particle types.**



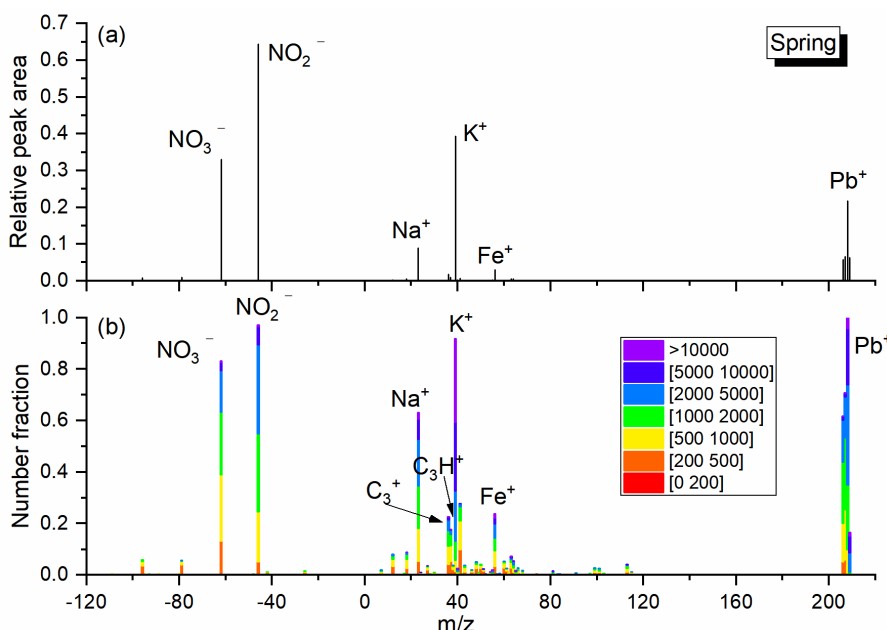

**Figure 2. The average mass spectrum and digital spectrum of lead nitrate particles during spring.**



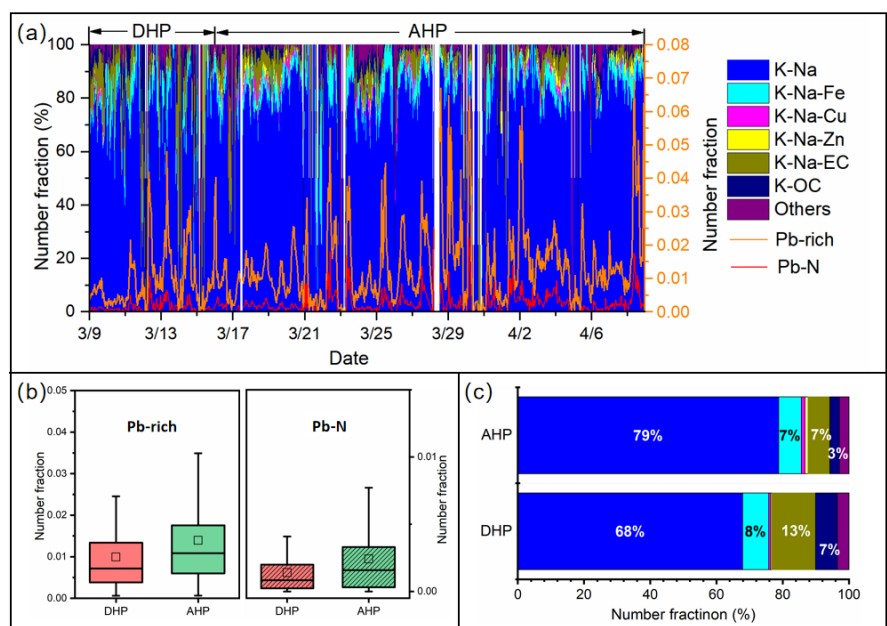


**Figure 3. The time series of the main Pb-rich particle type particles (a), the number fractions of Pb-rich and Pb-N particles (b), and the relative abundance of seven types particles (c) during (DHP) and after the heating period (AHP) in spring.**



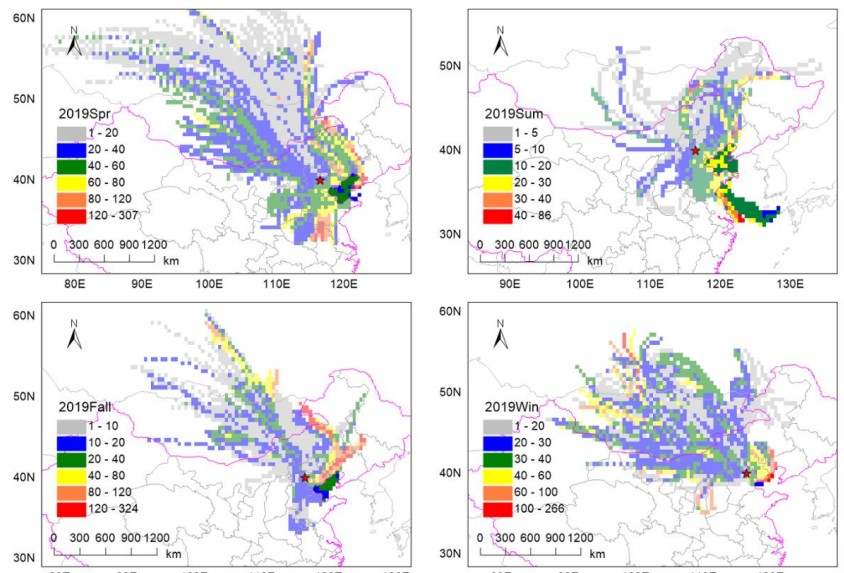

**Figure 4. CWT plots of potential source areas of Pb-rich particles from spring to winter, respectively.**



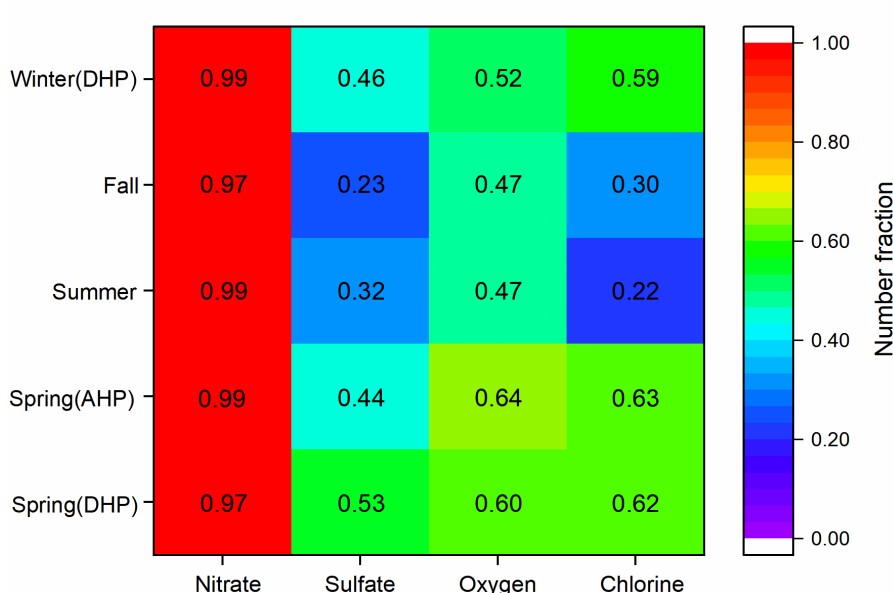

**Figure 5. Number fraction of nitrate ($m/z$ −46 [NO$_2$]$^-$ or −62 [NO$_3$]$^-$), sulfate ($m/z$ −97 [HSO$_4$]$^-$), oxygen ($m/z$ −16 [O]$^-$ or −17 [OH]$^-$), and chlorine (m/z −35 [$^{35}$Cl]$^-$ or −37 [$^{37}$Cl]$^-$) in Pb-rich particles from spring to winter, respectively.**



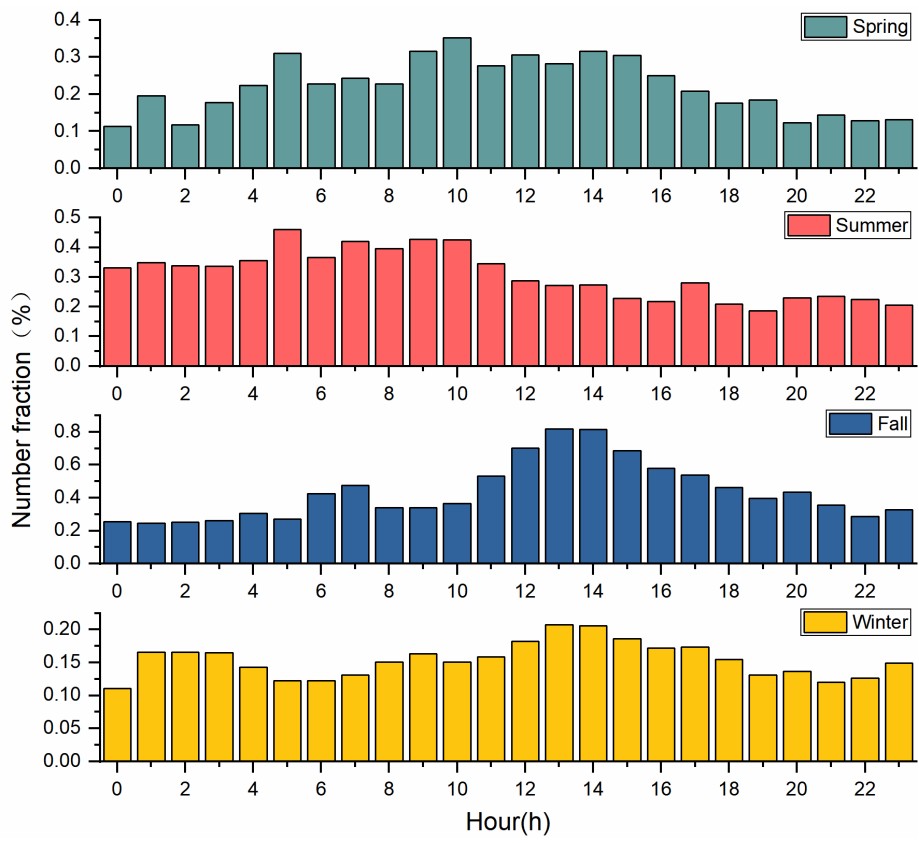

**Figure 6. The diurnal variation of the number fractions of Pb-N particles in all the detected particles from spring to winter, respectively.**




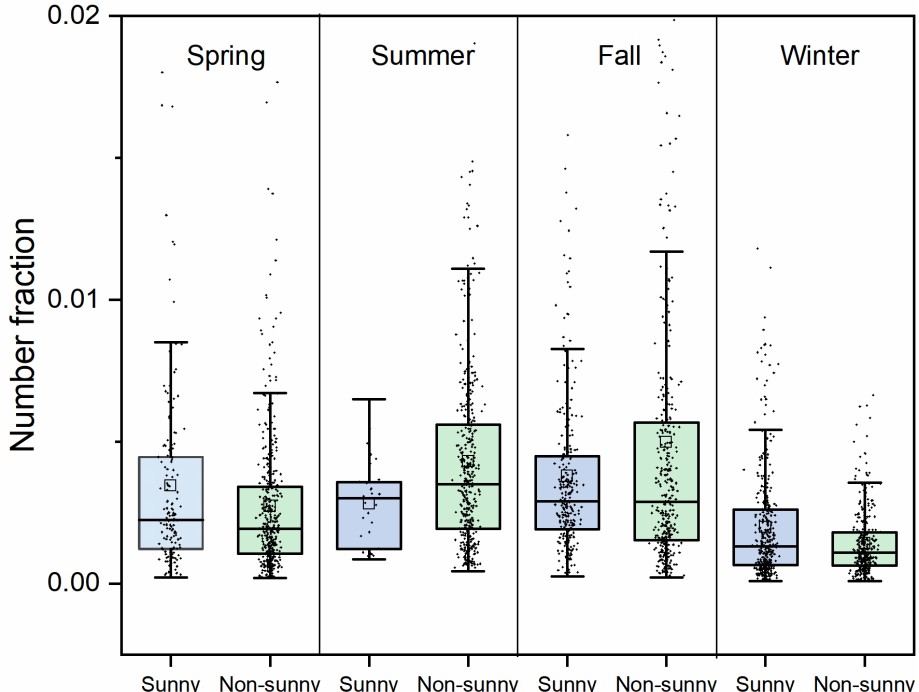

**Figure 7. The number fractions of Pb-N particles in Pb-rich particles on sunny and non-sunny days during four**
**seasons.**



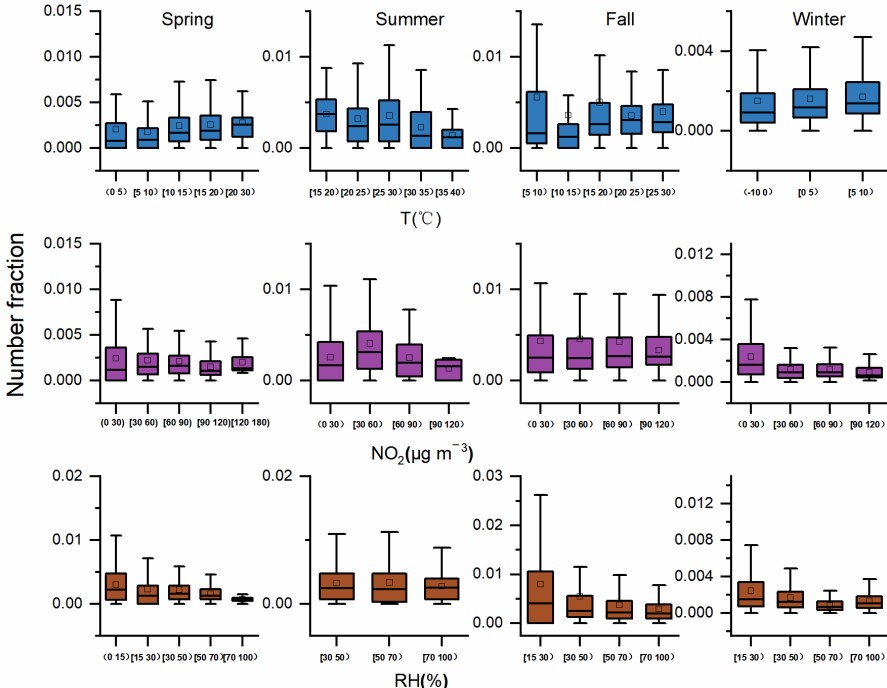

**Figure 8. The relationship between T, NO₂, RH and number fractions of Pb-N particles in Pb-rich particles from spring to winter, respectively.**



**Table 1 A summary of dates of SPAMS measurements, number and number fraction (Nf) of the detected Pb-rich particles.**

| species | Spring | | Summer | | Fall | | Winter 2019 | |
|---|---|---|---|---|---|---|---|---|
| | Number | Nf(%) | Number | Nf(%) | Number | Nf(%) | Number | Nf(%) |
| All particles[a] | 2058001 | | 654094 | | 1459253 | | 3430337 | |
| Pb-containing | 56738 | 2.8%[b] | 18893 | 2.9%[b] | 43763 | 3.0%[b] | 149590 | 4.4%[b] |
| Pb-rich | 23261 | 1.2%[b] | 8034 | 1.2%[b] | 17930 | 1.2%[b] | 23941 | 0.7%[b] |
| | | 44.5%[c] | | 42.5%[c] | | 40.9%[c] | | 16.0%[c] |
| K-Na | 20004 | 79.2%[d] | 6769 | 84.3%[d] | 14996 | 83.6%[d] | 18779 | 78.4%[d] |
| K-Na-Fe | 1725 | 6.8%[d] | 535 | 6.7%[d] | 879 | 4.9%[d] | 1270 | 5.3%[d] |
| K-Na-Cu | 297 | 1.2%[d] | 236 | 2.9%[d] | 598 | 3.3%[d] | 382 | 1.6%[d] |
| K-Na-Zn | 231 | 0.9%[d] | 7 | 0.1%[d] | 32 | 0.2%[d] | 86 | 0.4%[d] |
| K-Na-EC | 1594 | 6.3%[d] | 358 | 4.5%[d] | 263 | 1.5%[d] | 2128 | 8.9%[d] |
| K-OC | 696 | 2.8%[d] | 77 | 1.0%[d] | 783 | 4.4%[d] | 989 | 4.1%[d] |
| Others | 714 | 2.8%[d] | 52 | 0.6%[d] | 379 | 2.1%[d] | 307 | 1.3%[d] |
| Pb-N | 4314 | 17.1%[e] | 2292 | 28.5% | 7015 | 39.1% | 4712 | 19.7% |

[a] All particles were particles with both positive and negative spectra.
[b] Number fraction (Nf) was calculated through the number of Pb-containing or Pb-rich particles by the number of all particles over the sampling period.
[c] Number fraction (Nf) was calculated through the number of Pb-rich particles types by Pb-rich particles over the sampling period.
[d] Number fraction (Nf) was calculated through the number of Pb-rich particles by Pb-containing particles over the sampling period.
[e] Number fraction (Nf) was calculated through the number of Pb-N particles by Pb-rich particles over the sampling period.


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
