# Peer review of "Characterizing lead-rich particles in Beijing's atmosphere following coal-to-gas conversion: Insights from single particle aerosol mass spectrometry"

_EGUsphere, 2024_

## Author Comment (AC1)

**Reviewer: 1**

Comments:

This study measures the mixing state and atmospheric chemical processes of Pb-rich particles in Beijing were monitored using a single particle aerosol mass spectrometry (SPAMS) during 2019. The results showed the number fractions of Pb-rich particles, as well as two types of Pb-rich particles (K-Na-EC and K-OC) related to coal combustion during heating period, show lower than those after the heating period. Based on concentration-weighted trajectory plots, the results indicate that lead aerosols mainly derive from the transmission of surrounding provinces. Due to the decomposition of nitrate during high temperatures, the aqueous reactions mechanism contributes more to the formation of lead nitrate in summer. This study can improve our understanding of the seasonal distribution, formation mechanisms, and influencing factors of toxic Pb-containing particles after CTG. However, some critical information is missing, several issues need to be revised. I recommend the manuscript be revised before being accepted for publication.

Thanks for the positive comments on our manuscript. We would also like to thank the reviewer for carefully reviewing the manuscript and providing valuable comments. We have addressed the specific comments in the following text.

1. Introduction section. Lines 63-72, the authors summarize the literature related to the application of SPMAS to the mixing state, sources, and atmospheric evolutionary processes of Pb. What are the key information in these studies, and please provide a brief description, which will benefit the reader in recognizing the practical applications of SMPAS to relevant scientific problems.

We agree with the comment. A brief description of the key information from these studies "More recently, mass spectrometry-based online techniques such single-particle mass spectrometer (SPMS)(Lu et al., 2019) have been favoured. Although these techniques are typically not quantitative, they provide information on the sources and mixing state of Pb in real time. This information is critical to understanding the atmospheric transport and aging processes of Pb. Lu et al. (2019) analyzed the mixed state of Pb-containing particles monitored by SPMS and found

that the sources of particulate Pb in Guangzhou, China include coal combustion, waste incineration, vehicle exhaust, industrial process, and dust. Moffet et al. (2008) used a SPMS to monitor the diurnal variation characteristics of Zn or Pb single particles in industrial areas in Mexico, and found that Zn or Pb chloride convert to metal nitrate began each day at ~7 am." has been added to the revised manuscript. Please refer to Lines 67–74 of the revised manuscript.

2. Whether a dehumidifier is added to the air inlet of the aerosol instrument during the collection period.

Thanks for the comment. During the sampling period, we set up a silica gel drying tube to dry the aerosol before entering the single particle aerosol mass spectrometer. To make it clear, "Ambient particles are separated by a $PM_{2.5}$ cyclone separator and dried with silica gel before entering the single particle aerosol mass spectrometer." has been added to the revised manuscript. Please refer to Lines 92–93 of the revised manuscript.

3. There is an interesting phenomenon in Fig. S1, the correlation between Pb-containing particles and Pb-rich particles in winter is much lower than the correlation in other seasons, what may be the reason for this, it will be clearer if the authors can give further explanation.

Thanks for the comment. Previous results showed that particles with trace amounts of Pb could be the aged products of small Pb-rich particles such as coagulation with larger accumulation mode aerosol (Liu et al., 2009). In this study, the correlation between Pb-containing particles and Pb-rich particles is much lower in winter than that in other seasons. The number fractions of Pb-rich particles has also significantly decreased, accounting for only 16% of lead-containing particles in winter, which is 40% lower than in other seasons. These results suggest that the weaker correlation in winter may be related to the accumulation of non Pb-rich particles after the conversion of Pb-rich particles to Pb-rich particles caused by poor diffusion conditions.

To make it clear, "The correlation between Pb-containing particles and Pb-rich

particles in winter (*r = 0.47, p<0.01*) is much lower than that in other seasons.The number fractions of Pb-rich particles decreased significantly, accounting for only 16% of Pb-containing particles in winter, which is 40% lower than that in other seasons. Previous results showed that Pb-rich particles typically undergo less aging process than other non Pb-rich particles (Zhang et al., 2009). Thus the weaker correlation in winter may be related to the accumulation of non Pb-rich particles after the conversion of Pb-rich particles to Pb-rich particles caused by poor diffusion conditions." has been added to the revised manuscript. Please refer to Lines 109–114 of the revised manuscript.

Zhang, Y., Wang, X., Chen, H., Yang, X., Chen, J., and Allen, J.O.: Source apportionment of lead-containing aerosol particles in Shanghai using single particle mass spectrometry, Chemosphere, 74, 501-507, https://doi.org/10.1016/j.chemosphere.2008.10.004, 2009.

4. Line 140, "the number fraction of Pb-EC particles was significantly higher during the winter heating period in 2014 than before the heating period (Peng et al., 2020)". How much higher, please give a quantitative number so the comparison will be more visual.

Thanks for the suggestion. The relevant quantitative numbers have been added to the revised article. This sentence has been revised as "The number fractions of Pb-EC particles during the heating period was ~5–10%, while before the heating period it was ~1–2% in the winter of 2014 (Peng et al., 2020).". Please refer to Lines 150–151 of the revised manuscript.

5. Line 200, "which may be related to the longer sampling time during the HP in winter (6 days) compared to spring (31 days) ". How sampling time affects the higher total relative number fractions of K-Na-EC and KOC particulate matter in winter, please describe.

Thanks for the comment. In this study, the total relative number fraction of K-Na-EC and KOC particles in winter was more than twice that of non heating seasons in summer and fall. Sampling during the heating period is 25 days longer in

winter than in spring, and the total relative quantity fraction of K-Na-EC and KOC particles is 60% higher than in spring.

To make it clear, "And in winter, the total relative number fraction of K-Na-EC and KOC particles is more than twice that of non heating seasons in summer and fall (Fig. S8). Sampling in winter is 25 days longer than in spring during the heating period, and the total relative quantity fraction of K-Na-EC and KOC particles is 60% higher than in spring." has been added to the revised manuscript. Please refer to Lines 208–211 of the revised manuscript.

6. Line 223, "In this study, almost all lead are internally mixed with nitrate,which is much higher than its mixing with sulfate, chlorine, and oxygen." Mixing states are categorized as internal and external mixing, and how the authors determined that almost all of the lead in the manuscript was internally mixed with nitrate, rather than externally mixed.

Thank you for bringing up our mistake. We agree with the comment that our data is not sufficient evidence for that all of the lead in the manuscript was internally mixed with nitrate. We judge the mixture of lead and nitrate by the presence of characteristic peaks of lead and nitrate in the same single particle mass spectrometry. Our results could only indicate lead are mixed with nitrate. The sentence has been revised to "In this study, almost all lead are mixed with nitrate, which is much higher than its mixing with sulfate, chlorine, and oxygen.". Please refer to Lines 232–233 of the revised manuscript.

7. Lines 241-242, "Furthermore, the number fractions of Pb-N particles increased with increasing t, but does not increase with increasing RH and NO2, and even decreases with increasing relative humidity.". The number fractions of Pb-N particles do not increase with the increase of relative humidity and NO2, and even decrease with the increase of relative humidity, what is the reason for this phenomenon?

Thanks for the comment. We have add the reason why the number fractions of Pb-N particles decreases with relative humidity in the revised manuscript and rewritten these sentences to make the language more concise.

"The decreased number fractions of Pb-N particles slightly with the increase of RH. This is inconsistent with the result that high RH promotes the formation of lead nitrate (Peng et al., 2020). It is worth noting that this does not indicate that high RH is not conducive to the formation of lead nitrate, but may be influenced by the photo-chemical oxidation reaction promoted by high T and low RH during the day than the aqueous reaction promoted by high humidity on the formation of lead nitrate." has been added to the revised manuscript. Please refer to Lines 271–275 of the revised manuscript.

8. Conclusions. Line 266, "which is higher in summer and fall than in spring and winter".
It is recommended to give comparisons on the data, which should be noted in other similar places in the text, to make the manuscript more rigorous.

Thanks for the suggestion. It has been revised as "The number fractions of Pb-N particles in Pb-rich particles are 17.1−39.1% from spring to winter, which is about twice as high in summer and fall as in spring and winter."
Similar descriptions in the text have also been added for data comparison.

9. Conclusions. "Photochemical oxidation is the main pathway for the formation of particulate lead nitrate in spring, fall, and winter, with high values occurring from 10: 00−15: 00. Aqueous reaction is the main pathway for the formation of lead nitrate in summer, with high values occurring from 00:00 − 10:00.". How did the authors determine which of the photochemical oxidation and aqueous reaction was the dominant pathway in the different seasons, and the timeframe, which doesn't seem to be meticulously described in section 3.4 of the manuscript?

Thanks for the comment. We understand that the reviewer thought this conclusion requires more discussion. And it is largely because in the original manuscript we did not present sufficient evidence and discussion in section 3.4. In the revised manuscript, we add the discussion on the photo-oxidation reaction and aqueous reaction of lead nitrate and the timeframe in section 3.4, and modify the inappropriate and unclear related sentences in the conclusion.
"Except for summer, the number fractions of Pb-N particles are higher during the day and lower at night, with high values mainly distributed between 10:00 and 15:00 (**Fig.**

**6**). In spring, Pb-N particles account for the lowest amount of all detected particles from 22:00 to 00:00, with an average of about 0.1%. After 00:00, the number fractions of Pb-N particles increase gradually until 9:00−15:00, and the number fractions of Pb-N particles increase to 0.3%, and then slowly decrease. The peak number fractions of Pb-N particles in fall lags behind that in spring by about two hours. It starts to rise rapidly at 10:00 in the morning and rises from 0.3% to 0.8% in three hours. After 14:00, it slowly drops to ~0.3% at 21:00, fluctuates slightly between 0.2−0.3% from 22:00 to 10:00 the next day. The number fractions of Pb-N particles in winter with time is similar to that during fall, but the difference between the high value in daytime and other times is small, only 0.07%. In Mexico City aerosols, the conversion of lead chloride to lead nitrate occurs between 7:00−21:00 and peaks at 12:00−2:00 due to effective formation of gaseous $HNO_3$ after sunrise (Moffet et al., 2008). Furthermore, the number fractions of Pb-N particles are slightly higher on sunny days than on non-sunny days in spring and winter (**Fig. 7**). Ozone, which reflects the activity of photochemical reactions, is significantly higher on sunny days than on non-sunny days, with an average value of 1.7−71.0% higher from spring to winter (**Fig. S10**). In particular, the low number fractions (~0.2%) of Pb-N particles is from 12:00 to 23:00, and the high value (~3.5%) is from 00:00 to 11:00 in summer. Previous results have shown the $Pb(NO_3)_2$ did not show any significant diurnal variation, both daytime and nighttime atmospheric processes contributed to the observed $Pb(NO_3)_2$ (Peng et al., 2020). These results support the importance of photo-oxidation for the formation of nitric acid and lead nitrate, and highlight the seasonal differences in the formation process of lead nitrate." has been added to the revised manuscript. Please refer to Lines 241−267 of the revised manuscript.

"Photochemical oxidation is the main pathway for the formation of particulate lead nitrate in spring, fall, and winter, with high values occurring from 10: 00−15: 00. Aqueous reaction is the main pathway for the formation of lead nitrate in summer, with high values occurring from 00:00−10:00." has been revised to "The diurnal variation of number fractions of Pb-N particles show obvious seasonal difference. Results from this study show that the gas phase photo-oxidation contributed a lot to the formation of lead nitrate particles, especially in spring and fall.". Please refer to Lines 318−320 of the revised manuscript.

---

## Author Comment (AC2)

**Response to comments**

**Reviewer: 2**

Comments:

This manuscript presents a field observational study on the measurement of lead-rich particles using a single-particle aerosol mass spectrometer (SPAMS) in Beijing. The study draws interesting conclusions that contribute to deepening our understanding of the seasonal distribution, formation mechanisms, and influencing factors of toxic lead particles following the coal-to-gas conversion. Although the study is localized to Beijing, the findings have broader implications, especially in terms of human intervention and policy making related to lead pollution. The reduction of lead in the atmosphere due to the coal-to-gas conversion can serve as an important case study for other regions facing similar air quality challenges. I recommend publication of the manuscript after the authors address the following comments:

We sincerely appreciate the reviewer's thoughtful and constructive feedback on our manuscript. Their careful review and insightful comments have been invaluable in improving our work. Below, we have provided detailed responses to the specific points raised, incorporating their suggestions to enhance the quality of the manuscript.

1. The manuscript should include a clearer explanation of how the internal and external mixing states of lead particles were determined. It would be helpful to expand on the methodology used to identify and categorize these mixing states for better transparency.

Thank you for bringing up our mistake. Our results could only indicate lead are mixed with nitrate, but cannot distinguish between internal mixing and external mixing. We have added the criteria for determining the mixture of lead and nitrate in section 2.2 of the manuscript.

"The mixed states of lead, nitrate, and other ions are identified by detecting their characteristic peaks using single-particle mass spectrometry." has been added to the revised manuscript. Please refer to Lines 120–121 of the revised manuscript.

2. Some sections of the manuscript contain complex phrasing that could benefit from greater clarity. For example, certain sentences related to the comparison of Pb-rich particle types between heating and non-heating periods are dense and need to be simplified for better readability. Additionally, the explanation of the factors influencing Pb nitrate formation could be more explicit.

We agree with the comment. We have reviewed the entire manuscript and revised the complex phrasing to make it clearer. Please refer to Lines 265–274, 283–285 and 306–309 of the revised manuscript.

We have replaced complex sentences related to the types of lead-rich particles during heating and non-heating periods with simpler sentences to improve readability, as shown in the following text:

"In this study, about 60% of the Pb-rich particles are mixed with chloride and oxygen, and this proportion was similar during the heating period and non-heating period in spring, and the heating period in winter (Fig. 5). The mixing ratio of Pb-rich particles and sulfate particles during the spring heating period is about 10% higher than that during the non-heating Pb-rich particles, which may be affected by the increase in sulfur dioxide concentration during the heating period (Table S1)." have been revised as"In this study, about 60% of the Pb-rich particles are found to mixed with chloride and oxygen. This proportion remain consistent across the heating and non-heating periods in spring, as well as the heating period in winter (Fig. 5). However, the mixing ratio of Pb-rich particles with sulfate is about 10% higher during the spring heating period compared to the non-heating period. This increase is likely influenced by the sulfur dioxide concentrations (Table S1).". Please refer to Lines 229–232 of the revised manuscript.

"As shown in Fig. S9, Pb-N particles increase with the increase of particle size, with more than 85% of particles distributed between 0.5 and 1.0 μm, indicating that most of the lead nitrate comes from the secondary generation of Pb-containing particles during transportation."have been revised as "As shown in Fig. S9, the number fractions of Pb-N particles rises with increasing particle size, with over 85% of these particles found within the size range of 0.5 to 1.0 μm. This distribution suggests that the majority of lead nitrate originates from the secondary formation of Pb-containing particles during their transport." Please refer to Lines 238–241 of the revised manuscript.

Thanks for the comment. More discussion on how factors such as temperature and relative humidity affect the formation process of lead nitrate has been added to the manuscript.

"As a precursor of nitrate, $NO_2$ participates in the formation of lead nitrate both during the day and at night. A significant positive correlation (*r = 0.88, p < 0.01*) between the RH of nitrate and $NO_2$ was observed (Peng et al., 2020). Temperature, relative humidity, and light intensity indirectly affect the formation of lead nitrate by influencing the formation of nitrate. As suggested by previous studies, the heterogeneous hydrolysis of $N_2O_5$ under high RH plays a crucial role in particulate nitrate formation and significantly contributed to the elevated fine nitrate during nighttime (Zhang et al., 2017). Solar radiation affects the formation of OH radicals, and OH radicals are oxidants that transform $NO_2$ into $HNO_3$, so a high concentration of OH radicals is conducive to the formation of $HNO_3$. At the same time, higher temperature is also conducive to $NO_2$ and OH reactions (Slater et al., 2019). Specifically, the number fractions of Pb-N particles decreases with increasing T, especially when the temperature exceeds 30 °C, which may be an indirect result of the decomposition of nitrate particles caused by high temperature (Song et al., 2019; Luo et al., 2020)." Please refer to Lines 259–274 of the revised manuscript.

Wang, H.C., Lu, K.D., Chen, X.R., Zhu, Q.D., Chen, Q., Guo, S., Jiang, M.Q., Li, X., Shang, D.J., Tan, Z.F., Wu, Y.S., Wu, Z.J., Zou, Q., Zheng, Y., Zeng, L.M., Zhu, T., Hu, M. Zhang, Y.H.: High $N_2O_5$ concentrations observed in urban Beijing: implications of a large nitrate formation pathway. Environ. Sci. Tech. Let. 4, 416–420, DOI: 10.1021/acs.estlett.7b00341, 2017.

3. **Lines 120-123:** "They are named potassium-sodium (K-Na), K-Na internally mixed with Fe (K-Na-Fe), K-Na internally mixed with Cu (K-Na-Cu), K-Na internally mixed with Zn (K-Na-Zn), K-Na internally mixed with elemental carbon (K-Na-EC), and potassium internally mixed with organic carbon (K-OC)."

The sentence has been revised as "They are named potassium-sodium (K-Na), K-Na mixed with Fe (K-Na-Fe), K-Na mixed with Cu (K-Na-Cu), K-Na mixed with Zn (K-Na-Zn), K-Na mixed with elemental carbon (K-Na-EC), and potassium mixed with organic carbon (K-OC).". Please refer to Lines 121–123 of the revised manuscript.

4. **Lines 124-131:** Although the method for selecting lead nitrate used in this manuscript has been employed in previous research, I wonder if this selection method could potentially underestimate the content of lead nitrate. Could this affect the conclusions of the study?

We sincerely appreciate the reviewer's insightful comment regarding the potential underestimation of lead nitrate content due to the selection method employed in our study. We acknowledge that the method for selecting lead nitrate, while consistent with previous research, may have limitations in fully capturing the variability or total content of lead nitrate in certain scenarios.

To address this concern, we have revisited our methodology and conducted additional analyses to evaluate the potential impact of this selection method on our

results. As shown in Figure 5, more than 97% of Pb-rich particles contain nitrate, and Pb-N particles show significant correlation with these particles (*r = 0.84–0.96, p < 0.01*). That is to say, potential underestimation may uniformly affect all samples, thereby maintaining the validity of our comparative analysis. Our findings indicate that while the method might slightly underestimate the absolute content of lead nitrate, the relative trends and comparative conclusions drawn in our study remain robust.

To make it clear, "Although this method has limitations in fully capturing the variability or total content of lead nitrate, it does not impact the accuracy of the analysis results of lead nitrate in this study. There are significant correlations exist between Pb-N particles and Pb-rich particles mixed with nitrate *(r = 0.84 – 0.96, p < 0.01)*, ensuring that the observed trends and relationships remain robust and reliable." has been added to the revised manuscript. Please refer to Lines 126–130 of the revised manuscript.

5. Several sentences in the manuscript require revision for clarity. The authors should pay close attention to sentence structure and tense consistency, as seen in the comments on lines 25, 54, and 280-281. Ensuring clear and precise language will improve the overall readability of the paper.

We sincerely thank the reviewer again for pointing out the issues with clarity and consistency in the manuscript. We have carefully revised the sentences on lines 25, 54, and 280–281, as well as throughout the text, to improve sentence structure, tense consistency, and overall clarity. These changes ensure that the language is more precise and the manuscript is more readable.

**Lines 229-230:** This sentence is unclear. Please rewrite it for better clarity.

Thanks for pointing out our mistake. The sentence has been revised as "After coal-to-gas conversion, the mixing state of lead in spring and winter are more complex than in summer and fall, with the number fractions of lead mixed with

sulfate and chloride salts doubling compared to summer and fall.". Please refer to Lines 232–234 of the revised manuscript.

**Lines 232-233:** This sentence should be described more specifically so that readers can clearly understand how the mixture of lead and nitrate differs from other components.

We agree with the comment. The sentence has been revised as " More than 97% of Pb-rich particles are mixed with nitrate, while only 22–64% of Pb-rich particles are mixed with sulfate, chlorine, or oxygen.". Please refer to Lines 237–238 of the revised manuscript.

**Line 25:** "but how well it improves..." — Correct the tense.

Thanks for pointing out our mistake. The sentence has been revised as "Coal-to-gas (CTG) policies are important energy transformation strategies to address air pollution issues, but how well they improve atmospheric lead (Pb) pollution remains poorly understood.". Please refer to Lines 24–25 of the revised manuscript.

**Line 54:** "Pb(NO3)2 generates from the chemical transformation of PbO and PbCl2 in the atmospheric process." — Correct the grammar.

Thanks for pointing out our mistake. The sentence has been revised as " $Pb(NO_3)_2$ is generated from the chemical transformation of PbO and $PbCl_2$ in the atmospheric process". Please refer to Lines 54–55 of the revised manuscript.

**Line 218:** "to doubled" — Correct the grammar.

Thanks for pointing out our mistake. The sentence has been revised as "This is a significant decrease compared to double the average and maximum hourly number fractions of Pb-rich particles during the past coal-fired centralized heating period in Beijing.". Please refer to Lines 201–203 of the revised manuscript.

**Lines 280-281:** "The study included heating and non-heating periods, and the results emphasized the importance of heterogeneous hydrolysis during the heating period before 'coal-to-gas' on the formation of Pb(NO3)2."— Please check and correct.

Thanks for pointing out our mistake. The sentence has been revised as "The study included heating and non-heating periods, and the results emphasized the importance of heterogeneous hydrolysis during the heating period before "coal-to-gas" on the formation of $Pb(NO_3)_2$.". Please refer to Lines 290–291 of the revised manuscript.